# Bose–Einstein Correlations in pp and pPb Collisions at LHCb [†]

**Bartosz Malecki** [‡] 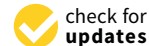

Institute of Nuclear Physics Polish Academy of Sciences, PL-31342 Krakow, Poland; Bartosz.Malecki@ifj.edu.pl
† Presented at the 18. Zimanyi School, Budapest 2018.
‡ On behalf of the LHCb Collaboration.

**Abstract:** Bose–Einstein correlations for same-sign charged pions from proton–proton collisions at $\sqrt{s} = 7$ TeV are studied by the Large Hadron Collider beauty (LHCb) experiment. Correlation radii and chaoticity parameters are determined for different regions of charged-particle multiplicity using a double-ratio technique and a Levy parametrization of the correlation function. The correlation radius increases with the charged-particle multiplicity, while the chaoticity parameter decreases, which is consistent with observations from other experiments. A similar study for proton-lead collisions at $\sqrt{s_{NN}} = 5.02$ TeV is proposed. These results can give valuable input for the theoretical models that describe the evolution of the particle source, probing both its potential dependence on pseudorapidity region and differences between proton–proton and proton–lead systems.

**Keywords:** femtoscopy; small systems; Bose-Einstein correlations; HBT

**PACS:** 13.87.Ce; 14.40.Aq

## 1. Introduction

Multi-particle production is a basic process in the field of high energy physics, yet it still lacks a satisfactory description. One of the interesting aspects of this phenomenon is the evolution of the particle source. Intensity interferometry, also known as Hanbury–Brown and Twiss interferometry (HBT) [1] is a very useful tool that can provide information on spatiotemporal structure of this region. This method allows one to observe effects of quantum correlations between same-sign charged hadrons of a given species emitted from a single particle source. Such correlations emerge from the quantum statistics describing the particular particle system. For bosons, Bose–Einstein correlations (BEC) are observed, which are manifested as an enhancement in the production of identical bosons with small relative difference in four-momenta. For fermions, the effect is the opposite and it is called Fermi–Dirac correlations (FDC). Using HBT interferometry, a correlation radius $R$ and a chaoticity parameter $\lambda$ can be determined. The correlation radius is related to the size of the particle source at freeze-out, while the chaoticity parameter contains information on the coherence of the particle emission.

Comparing results from analyses of the Bose–Einstein correlations in proton–proton (pp) and proton–lead (pPb) collisions is of particular interest. It provides extraordinary input for the development of theoretical models that aim to describe the process of particle production. Some of those models predict contradictory behavior in the size of the particle source in pp and pPb collisions. For example, the hydrodynamical model [2,3] states that the source should have a significantly bigger size in the case of a pPb system, while according to the model based on gluon saturation, this size should be very similar [4,5].

Bose–Einstein correlations were first observed in the field of elementary particle physics in 1959 [6] (Fermi–Dirac correlations slightly later) and were studied in many different collision systems and

energies since then. Among others, a number of analyses were conducted for $e^+e^-$ collisions at LEP [7–14] and for pp collisions at LHC by collaborations: ALICE, ATLAS and CMS [15–19]. BEC measurements were also done for heavy ion collisions, e.g., by ALICE [20] and STAR [21]. The pPb system has become of great interest recently, with results from ALICE [22], ATLAS [23] and CMS [24].

Studies of the BEC and FDC effects showed that the correlation parameters depend on a number of factors. It was already indicated by the results from LEP that the correlation radius is related to the mass (species) of the produced hadron—the heavier it is, the smaller the correlation radius. There are several theoretical models that aim to explain this phenomenon [25,26], but the current measurements cannot discard any of them. The correlation effects can be also studied in two or three dimensions—then one can obtain information on the correlation radii in two or three directions, respectively, which gives a more detailed description of the particle source shape. Results from multidimensional analyses of the correlation effects indicate that the source propagates mostly in the beam direction.

Analyses of the BEC effect for different colliding systems and energies showed that $R$ increases with the charged-particle multiplicity ($N_{ch}$) of an event [27], while $\lambda$ becomes smaller in most cases. Furthermore, the correlation radius rises approximately linearly with a cube root of the local $N_{ch}$ density in pseudorapidity $< \mathrm{d}N_{ch}/\mathrm{d}\eta >^{1/3}$. The scaling factor in this relation changes with the type of colliding system [22]. It was also observed that the correlation radius is almost independent of the collision energy, when comparing results for similar $N_{ch}$ values [15,17,19]. Current measurements show that $R$ and $\lambda$ decrease with the rising average transverse momentum of the particles in a pair ($k_T$). Many analyses try to determine how the parameters of correlation are related to the factors mentioned and if some scaling behavior can be observed.

The aim of this paper is to present results from the study of the BEC effect for pairs of same-sign charged pions from high-energy pp collisions recorded by the Large Hadron Collider beauty (LHCb) experiment [28]. A one-dimensional analysis is performed and the correlation parameters $R$ and $\lambda$ are determined for different regions of charged-particle multiplicity. The LHCb detector [29,30] has a unique acceptance ($2.0 < \eta < 5.0$) among other LHC experiments. Thus, the results for the forward direction are the first of their kind and are complementary to observations from other experiments that cover the central acceptance region. It allows one to study the potential dependence of the correlation parameters on pseudorapidity and gives an additional input to understand the process of particle production. A similar study of the BEC effect is ongoing at LHCb for pPb collisions. A direct comparison to the LHCb results on BEC effect in pp collisions will give a strong reference to establish whether the particle production process is different in those two systems. It will also allow one to verify the theoretical models and put additional constraints on their initial parameters (such as the initial transverse size of the colliding system).

## 2. Materials and Methods

### 2.1. LHCb Detector

The LHCb detector [29,30] is a single-arm spectrometer (see Figure 1) and covers the pseudorapidity range of $2.0 < \eta < 5.0$, which is a region unique among other experiments at the LHC. The most distinctive features of the LHCb detector are: A very precise system of track and primary vertex (PV) reconstruction, as well as its outstanding particle identification capabilities. The track reconstruction system consists of a vertex detector VELO (which is a silicon-strip detector, surrounding the interaction region), a station of silicon-strip detectors located right before a magnet, the magnet with a bending power of 4 Tm and three stations of silicon-strip and straw drift tubes placed downstream the magnet. The track reconstruction system is capable of determining particles' momenta with uncertainty between 0.5% (for low momenta) and 1% (for momenta at the level of 200 GeV/$c$). Particle identification is based on information from two Ring Imaging Cherenkov detectors (RICH), a system of calorimeters and muon stations. The RICH detectors are especially important for the BEC analysis, since they allow one to identify species of the charged hadrons.

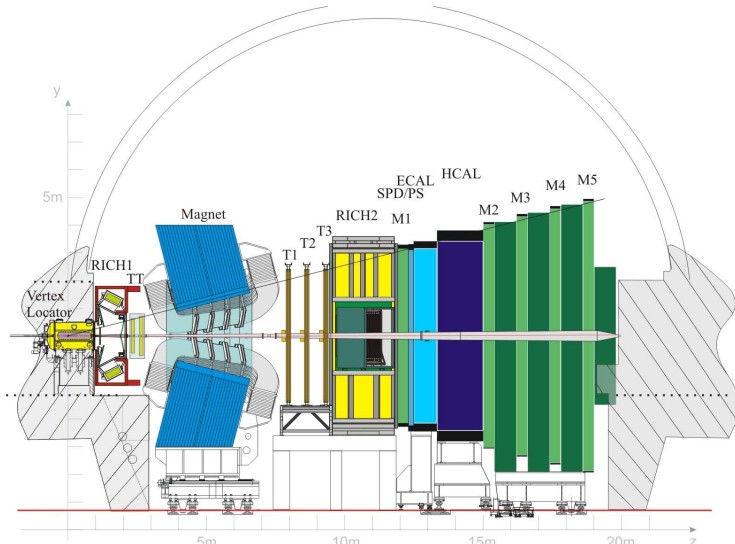

**Figure 1.** A schematic view of the LHCb detector in y-z plane. The z-axis is the beam axis and the x-axis is perpendicular to the page surface (figure from http://cds.cern.ch/record/1087860; 25 March 2019).

### 2.2. Data Sample

The data sample used in this study was collected by the LHCb experiment in 2011 from pp collisions at $\sqrt{s} = 7$ TeV centre-of-mass energy and contains $4 \times 10^7$ minimum bias events. A corresponding Monte Carlo (MC) sample of $2 \times 10^7$ events was prepared with the LHCb software [31] using PYTHIA 8 [32] as the event generator and GEANT4 [33] for the full detector simulation. Futher data analysis is performed within the ROOT framework [34]. The data is divided into three activity classes based on the VELO track multiplicity distribution (see Figure 2), which is a good approximation of the total $N_{ch}$ in an event. The low activity class contains a fraction of 48% PVs with the lowest multiplicities, the medium one corresponds to 37% PVs with higher multiplicities and the high activity class consists of the 15% PVs with highest multiplicities. The VELO track multiplicity is unfolded to the true $N_{ch}$ values using the simulation. The exact ranges of the charged-particle multiplicity regions for each activity class are showed in the Table 1.

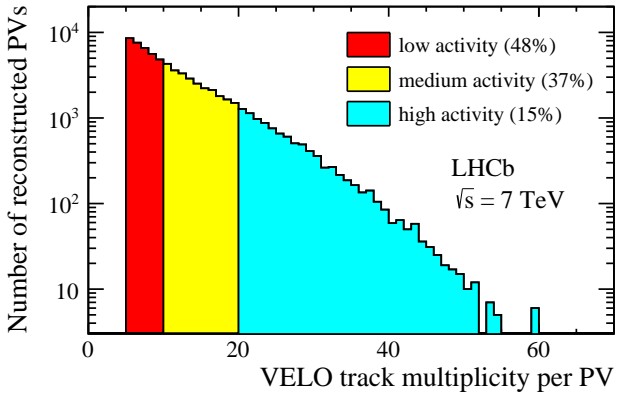

**Figure 2.** Multiplicity of the reconstructed VELO tracks assigned to a PV for the data. Different colors indicate three activity classes defined as fractions of the full distribution. The Figure is taken from [28].

### 2.3. Analysis Method

Study of the Bose–Einstein or Fermi–Dirac correlations is usually done using a Lorentz-invariant variable $Q = \sqrt{-(q_1 - q_2)^2}$, which is a measure of proximity in the four-momenta phase space for two particles with four-momenta $q_1$ and $q_2$ [35]. Experimentally, the correlation function is defined as:

$$C_2(Q) = N(Q)^{SAME}/N(Q)^{REF}, \tag{1}$$

where $N(Q)^{SAME}$ is the $Q$ distribution of the signal pairs (same-sign charged pions originating from a single PV). The distribution $N(Q)^{REF}$ comes from the reference sample, which should contain all phenomena that are present in the signal pairs, except for the BEC effect. That includes, e.g., long-range correlations (emerging due to energy/momentum conservation law) or Coulomb interactions between the charged pions in their final states. There are several ways of constructing the reference sample, but none of them meets all the above requirements perfectly. In this analysis, the reference pairs are created by mixing particles from different events, where by definition the BEC effect is not present. However, in this way also other kinds of correlations are removed from the sample, such as the long-range ones.

In order to correct for the imperfections in construction of the reference sample, the so-called double-ratio $r_d(Q)$ is introduced. It is a ratio of the correlation functions obtained for data and simulation. In both cases, correlation function is created in exactly the same way, but the BEC effect is switched off in the simulation. Due to that, structures that originate from phenomena properly simulated in the MC sample, are removed from the initial correlation function for data and in an ideal case only the pure BEC signal should be visible. In this way, structures related to, e.g., the long-range correlations can be eliminated from the correlation function. The double-ratio is to a large extent insensitive to effects due to efficiency, detector occupancy and acceptance, as well as the choice of selection criteria.

Coulomb interactions between charged particles in their final states is one of the phenomena that are not present in the simulation. This effect can influence the shape of Q distributions. In the case of same-sign charged particles, a repulsive interaction leads to a decrease in the correlation function for small Q values. For particles with an opposite charge this effect is reversed. The correlation function for data is corrected for the Coulomb interaction effects by applying a Gamov penetration factor [36].

The correlation function is usually parametrized as a Fourier transform of the static source density distribution [37], $C_2(Q) = N(1 + \lambda \exp(-|RQ|^{\alpha_L}))$, where $R$ is the correlation radius, $\lambda$ denotes the chaoticity parameter and $N$ is a normalization factor. Parameter $\alpha_L$ is a Levy index of stability and corresponds to the assumed source density distribution. In this analysis, an exponential density distribution of a static particle source ($\alpha_L = 1$) is used, which leads to:

$$C_2(Q) = N(1 + \lambda \exp(-RQ)) \times (1 + \delta Q). \tag{2}$$

The factor $(1 + \delta Q)$ is not related to the BEC effect itself, but it's added to account for the long-range correlations that are manifested for the larger values of $Q$.

## 3. Results

Fits to the double-ratio distributions obtained for the three $N_{ch}$ regions are performed using the parametrization (2). An example of the fit for the middle activity class is shown in the Figure 3. A clear enhancement due to the BEC effect is seen for the values of Q approaching 0 GeV. The fit results are summarized in the Table 1. The systematic uncertainty (about 10% in each activity class) is dominated by the MC generator tunings and pile-up effects.

**Table 1.** Results of fits using parametrization (2) to the double-ratio for three different activity classes and the corresponding $N_{ch}$ bins. Statistical and systematic certainties are given separately (in this order). The Table is taken from [28].

| Activity | $N_{ch}$ | $R$ (fm) | $\lambda$ | $\delta$ (GeV$^{-1}$) |
|----------|----------|----------|-----------|-----------------------|
| Low | [8, 18] | $1.01 \pm 0.01 \pm 0.10$ | $0.72 \pm 0.01 \pm 0.05$ | $0.089 \pm 0.002 \pm 0.044$ |
| Medium | [19, 35] | $1.48 \pm 0.02 \pm 0.17$ | $0.63 \pm 0.01 \pm 0.05$ | $0.049 \pm 0.001 \pm 0.009$ |
| High | [36, 96] | $1.80 \pm 0.03 \pm 0.16$ | $0.57 \pm 0.01 \pm 0.03$ | $0.026 \pm 0.001 \pm 0.010$ |

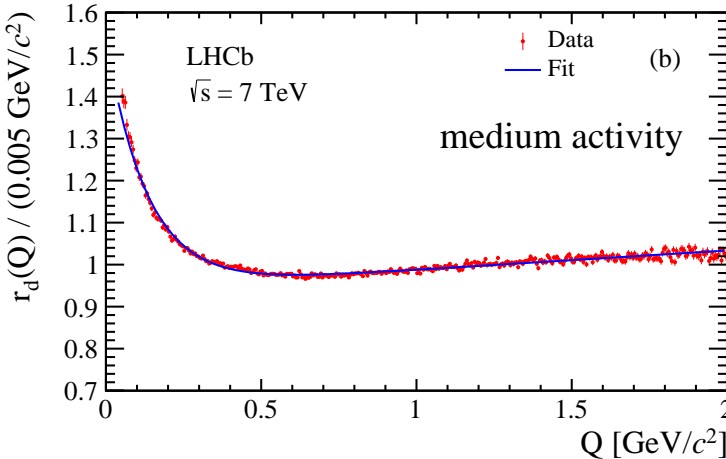

**Figure 3.** Results of the fit to double-ratio for same-sign charged pions from the middle activity class. The red points represent data with the statistical uncertainties, while the blue line denotes the fit using parametrization (2). The Figure is taken from [28].

The BEC parameters as a function of the activity classes are shown in the Figure 4. It is observed that $R$ increases with the charged-particle multiplicity, while $\lambda$ decreases, which is consistent with other observations from LEP and LHC. Using the simulation, the LHCb charged-particle multiplicity regions are extrapolated to the corresponding ones in the ATLAS acceptance and a comparison with the ATLAS results [17] is made. It is found that the BEC parameters in the forward direction are slightly lower than those at the central acceptance region. It could indicate that the correlation parameters depend on pseudorapidity. This effect will be studied in more detail by other analyses at LHCb, e.g., in the ongoing research for pPb collisions.

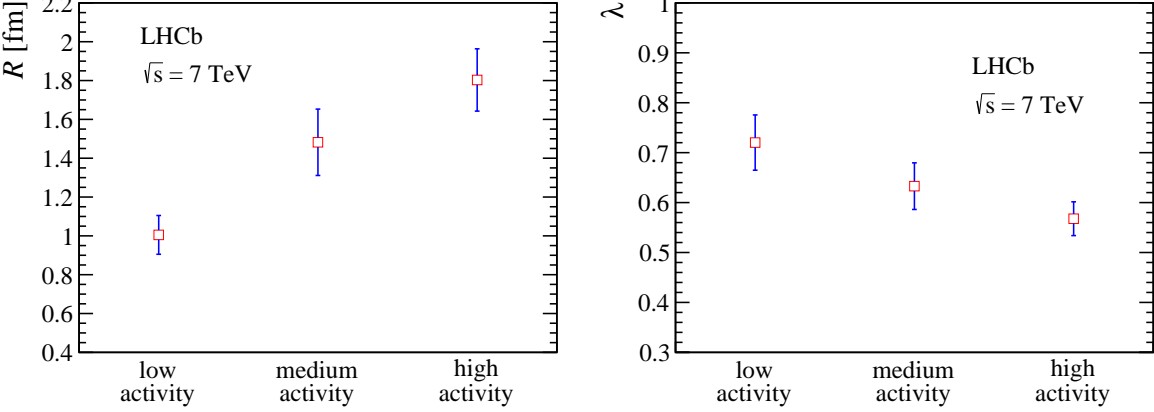

**Figure 4.** (**left**) Correlation radius $R$ and (**right**) chaoticity parameter $\lambda$ as a function of activity. Error bars indicate the sum in quadrature of the statistical and systematic uncertainties. The points are placed at the centres of the activity bins. The Figures are taken from [28].

## 4. Discussion

Bose–Einstein correlations for same-sign charged pions in pp collisions at $\sqrt{s} = 7$ TeV are studied by the LHCb experiment. The correlation radius $R$ and the chaoticity parameter $\lambda$ are determined for three different regions of the charged-particle multiplicity. It is observed that $R$ increases with $N_{ch}$, while $\lambda$ decreases, which is consistent with the previous observations from other experiments at, e.g., LEP and LHC. This measurement is the first of its kind in the forward region and shows the potential of LHCb in similar analyses. The unique acceptance of the LHCb detector among other LHC experiments allows one to obtain results that are complementary to the studies in the central

region of pseudorapidity. The LHCb measurement is compared with ATLAS [17] and the correlation parameters in the forward region seem to be slightly lower than those in the central acceptance region. This behavior will be studied in the future analyses at LHCb.

One of the advancing analyses at LHCb is the study of BEC correlations for same-sign charged pions in pPb collisions at $\sqrt{s_{NN}} = 5.02$ TeV centre-of-mass energy per nucleon. This measurement is carried out in different regions of both charged-particle multiplicity and average transverse momentum of the particles in a pair. This will allow one to compare the results to the other LHC experiments and study the potential dependence of the correlation parameters on pseudorapidity. Furthermore, a direct comparison between the LHCb results for both pp and pPb collisions will give insight into the differences in evolution of those two systems. This will provide additional constraints on the parameters of theoretical models that aim to describe the complex process of multi-particle production (such as the initial transverse size of the colliding system).

The usual approach in the BEC studies is to assume a static source of particles, which leads to the Levy-type parametrizations of the correlation function such as (2). One of the alternative strategies is to use a $\tau$-model [38], which accounts for the time evolution of the particle source. This model was proved to be successful in the description of the data in case of $e^+e^-$ collisions [12], however it has not been used for the pPb system yet. Apart from the interesting comparisons between central/forward acceptance regions and pp/pPb collisions, testing the $\tau$-model is one of the goals in the LHCb analysis of the BEC effect in pPb system.

There are many other interesting future directions of the BEC/FDC studies at LHCb, such as 3-body correlations (ongoing), a three-dimensional analysis for same-sign charged pions, or the BEC effect in lead–lead collisions. One of the most intriguing possibilities is to perform this kind of measurement for D mesons, which would be a completely new result in terms of the hadron species. Preliminary studies suggest that such an analysis should be possible with the LHCb data.

**Funding:** This research was funded by Narodowe Centrum Nauki grant number 2013/11/B/ST2/03829 and 2018/29/N/ST2/01641.

**Acknowledgments:** This study was partially performed using the PL-GRID infrastructure.

**Conflicts of Interest:** The authors declare no conflict of interest. The funders had no role in the design of the study; in the collection, analyses, or interpretation of data; in the writing of the manuscript, or in the decision to publish the results.

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
