# Peer review of "Bose–Einstein Correlations in pp and pPb Collisions at LHCb"

_universe, doi:10.3390/universe5040095_

Round 1

Reviewer 1 Report

This article discusses work presented on behalf of LHCb at the 18 Zimanyi School 2018, and serves largely as a summary of results from Ref. [28].  

An analysis of HBT correlations was performed on data from 7 TeV proton-proton collisions as measured by the LHCb Collaboration.  Specifically, charged pion HBT correlations were measured using a 1-dimensional Levy parameterization, with Pythia simulations used to remove other dynamical correlations except Coulomb interactions, which were removed using a Gamov penetration factor.

The main result is the increase of the 1D radius and a decrease of the chaoticity parameter with increasing event activity, and a larger overall radius compared to measurements from ATLAS (though the comparison is not shown).  

These results are most interesting because of the rapidity coverage of the LHCb detectors that complement that of the other LHC collaborations, which cover mostly smaller rapidity regions.  As such, these results represent valuable and interesting contributions to the field, and even more so the various future directions which are discussed near the end of the text.

The text is clear and well written, and the results of sufficient interest.  I recommend publication in its current form.

Author Response

I would like to thank the Reviewer for this nice comments. I'll proceed with the publication process.

Reviewer 2 Report

In the manuscript, preliminary results for Bose-Einstein correlations in small systems obtained with the LHCb detector are presented. Overall, the manuscript contains interesting results, since LHCb can access the forward rapidity region which is complementary to any other LHC experiment. Since there is still a lively debate in the community about potential collectivity in small systems, this study will add interesting new constraints. I only have two minor questions and recommend publication of the manuscript. 

1) I find it a bit confusing that the title mentions pp and pPb collisions, while only the results for pp collisions are shown. Since this is clarified in the abstract, I understand why the pPb collisions are mentioned in the title, still I would have expected to see results in an article with this title. 

2) line 139: the authors mention that their results suggest a rapidity dependence that was not predicted by theoretical models. Can you clarify what kind of model is referred to and if longitudinal structures were included in the respective calculation? 

Author Response

Point 1: I find it a bit confusing that the title mentions pp and pPb collisions, while only the results for pp collisions are shown. Since this is clarified in the abstract, I understand why the pPb collisions are mentioned in the title, still I would have expected to see results in an article with this title.

Response 1: It is due to the LHCb Collaboration policy to show only published / approved results. The analysis for pPb collisions is still ongoing and we can’t show them yet (they haven’t been presented at the Zimanyi School as well). The talk at Zimanyi School was aiming to describe in general the LHCb programme for BEC studies and the interesting physics outcome that can be reached. The current study direction is to compare the pPb and already published pp results, which is why both of those systems are referred to in the title.

Point 2: line 139: the authors mention that their results suggest a rapidity dependence that was not predicted by theoretical models. Can you clarify what kind of model is referred to and if longitudinal structures were included in the respective calculation?

Response 2: Thank you for this comment, the claim about theoretical models is too general and wasn’t our original goal to be put this way. Such a strong statement is removed in the revised version.

Our intention was to point out that the BEC studies are usually performed only in different regions of charged-particle multiplicity and average transverse momentum of the pair. Measuring a dependence of the BEC parameters on pseudorapidity would give a valuable input for the development of theoretical models. Up to now, only ATLAS presents BEC studies performed in different regions of rapidity for pPb collisions. They observe dependency of the BEC parameters on rapidity, which is related to the intrinsic asymmetry of the pPb system. A similar measurement from LHCb will give a complementary result and potential differences between forward / central acceptances will be studied.